# Theoretical Investigation of Electron–Ion Recombination Process of Mg-like Gold

Luyou Xie *, Wenliang He, Shengbo Niu, Jinglin Rui, Yulong Ma and Chenzhong Dong *

Key Laboratory of Atomic and Molecular Physics and Functional Materials of Gansu Province, College of Physics and Electronic Engineering, Northwest Normal University, Lanzhou 730070, China
* Correspondence: xiely@nwnu.edu.cn (L.X.); dongcz@nwnu.edu.cn (C.D.)

**Abstract:** The L-shell dielectronic and trielectronic recombinations of highly charged Mg-like gold ions ($Au^{67+}$) in the ground state $2s^2 2p^6 3s^2\,{}^1S_0$ have been studied systematically. The recombination cross-sections and rate coefficients are carefully calculated for $\Delta n = 1$ ($2s/2p \rightarrow 3l$) transitions using a flexible atomic code based on the relativistic configuration interaction method and considering the Breit and QED corrections. Detailed resonance energies and resonance strengths are presented for the stronger resonances of the LM$n$ ($n = 3$–12) series. It is found that the contributions of the trielectronic recombination to the total cross-section is about 13.75%, which cannot be neglected. For convenience of application, the plasma rate coefficients are also calculated and fitted to a semiempirical formula, and in the calculations, the contributions from the higher excited resonance groups $n \geq 13$ are evaluated by an extrapolation method, which is about 2.93% of the total rate coefficient.

**Keywords:** electron–ion recombination; cross-sections; rate coefficients; Mg-like gold





## 1. Introduction

Dielectronic recombination (*DR*) is a resonant electron–ion recombination process of known importance in astrophysical and fusion plasmas [1,2]. A complete *DR* process can be seen as a resonant two-step process, in which a free electron is captured by an ion under simultaneous excitation of a bound electron, and in the second step, the intermediate excited state decays radiatively [1–4]. The *DR* process is a dominant channel occurring in the collisions of electron with highly charged ions (HCI*s*) and is generally labeled by an inverse Auger process notation. For example, the LM$n$ *DR* indicates that one L-shell electron is excited during the capture of one free electron, resulting in a resonance-excited state with one electron in the M-shell and another electron in the $n$-shell ($n$ = M, N, O . . . ). As one important line formation mechanism in plasmas, the cross-sections of *DR* near resonance energies usually surpass those of other competing processes by orders of magnitude. It strongly influences the total recombination rate and thus the charge state balance, energy-level populations, as well as the radiative spectrum of hot plasmas [1,5]. Hence, accurate *DR* cross-sections and rate coefficients are needed for modeling astrophysical and fusion plasmas [6–8].

Gold is used as a good inertial confinement fusion (ICF) target material, and much attention has been paid to gold plasmas recently [9–13]. The *DR* processes at different ionization stages of gold have been investigated in recent years. Spies et al. [14] reported the first measurements on the radiative and dielectronic recombination of Li-like gold ions at the heavy-ion storage ring ESR and obtained absolute recombination rates of $\Delta n = 0$ ($2s \rightarrow 2p$) transitions with very good energy resolution. Schneider et al. [15] observed the LMM *DR* resonances of Ne-like gold ions in the electron beam ion trap (EBIT) at the Lawrence Livermore National Laboratory and found that the measured resonance strengths for LMM resonances with $2p^{-1}_{3/2}$ cores are in good agreement with theoretical predictions. Liu et al. [16] investigated the *DR* of Ne-like gold ions employing a flexible atomic code

(FAC) [17] based on the relativistic configuration interaction method and determined its influence on the ionization balance of high-temperature plasma. Yang et al. [18] performed detailed level-by-level calculations for the total *DR* rate coefficients of Ne-like Au$^{69+}$ ions in the ground state using a FAC. Xiong et al. [19] measured the KLL *DR* resonance strengths of Li-like to O-like gold ions with the Tokyo-EBIT and compared the results with the theoretical predictions, which showed good agreement. Iorga et al. [20] studied the degree of linear polarization in dielectronic recombination K$\alpha$ satellite lines in Li-like gold ions within a density matrix formalism based on relativistic atomic data computed via relativistic distorted wave approximation.

In this work, we will focus on the L-shell $\Delta n = 1$ ($2s/2p \rightarrow 3l$) *DR* process of Mg-like Au$^{67+}$ ions in the initial ground state. Accuracy calculations of the atomic structure of these ions have been carried out and collected spectral data have been collected by many researchers [21–24]. Konan et al. [21] calculated the energy levels, wavelengths, transition rates and weighted oscillator strengths for strong E1 transitions using the AUTOSTRUC-TURE code. Hamasha et al. [22] calculated the energy levels, oscillator strengths and transition rates for the multipole transitions (E1, E2, M1, M2) between the ground and the excited states $3l \rightarrow nl'$ ($n = 4, 5, 6, 7$). Vilkas et al. [23] reported the energy levels and lifetimes related to $\Delta n = 0$ ($n = 3$) transitions. Hamasha et al. [24] calculated the atomic structure and spectral data for E1 transitions with $\Delta n \neq 0$ ($n = 3 \rightarrow 4, 5, 6, 7$). To our knowledge, however, theoretical research on the *DR* processes of Mg-like Au$^{67+}$ ions is scare. Using a flexible atomic code [17] based on the relativistic configuration interaction (RCI) method, we studied the *DR* cross-sections and rate coefficients for inner *L*-shell $2s/2p$-core excitation in isolated resonance approximation. Beyond dominant dielectronic recombination, the higher-order trielectronic recombination is also discussed. In Section 2, the theoretical method is described. In Section 3, the resonant recombination cross-sections and rate coefficients are presented and discussed. Finally, some brief conclusions are given in Section 4.

## 2. Theoretical Methods

The inner L-shell $\Delta n = 1$ resonant recombination process of Mg-like Au$^{67+}$ ions in the ground state can be described as follows:

$$Au_i^{67+}\left[2s^2 2p^6 3s^2\right] + \varepsilon e^- \rightarrow Au_d^{66+**} \begin{cases} \left[(2s2p)^{-1}3s^2 3lnl'\right](DR)(a) \\ \left[(2s2p)^{-1}3s^{-1}3l^2 nl'\right](TR)(b) \end{cases} \rightarrow Au_f^{66+*}\left[3s^2 nl' + 3s3lnl' + 3l^2 nl'\right] + h\nu$$

$$\downarrow$$

$$Au_f^{67+*}\left[3s^2, 3snl, n = 3-5\right] + \varepsilon' e^- \tag{1}$$

where Equation (1) (*a*) describes the dielectronic recombination channels and Equation (1) (*b*) denotes the trielectronic recombination (*TR*) channels. The notation $(2s2p)^{-1}$ indicates a vacancy in the inner L-shell, corresponding to the excitation of either a $2s$ or $2p$ electron to the $3l$ ($l = p, d$) subshell for the $\Delta n = 1$ transition. Here and below, $n = 3$–12, and all their possible angular momentums $l'$ are included. Those resonances' double- and triple-excited states can also autoionize to the ground or to singly excited states of Mg-like Au$^{67+}$ ions.

In isolated resonance approximation [25], the *DR* strength of an individual resonance, involving its excitation from an initial state $i$ via a resonance-excited state $d$ to a final state $f$, can be given by the following equation [26] (atomic units):

$$S_{idf} = \int_0^\infty \sigma_{idf}(\varepsilon)d\varepsilon = \frac{\pi^2}{E_{id}} \frac{g_d}{2g_i} A_{di}^a B_{d,f}^r \tag{2}$$

where $\sigma_{idf}(\varepsilon)$ is the *DR* cross-section as a function of electron energy $\varepsilon$; $E_{id}$ denotes the energy separation of the resonance-excited state $d$ from the initial state $i$, namely, the resonant energy; $g_i$ and $g_d$ are the statistical weights of the states $i$ and $d$, respectively; $A_{di}^a$ is

the autoionization rate from the states $d$ to $i$; and $B_{d,f}^r$ the radiative branching ratio, written as follows:

$$B_{d,f}^r = \frac{A_{df}^r}{\sum\limits_{f'} A_{df'}^r + \sum\limits_{i'} A_{di'}^a} \tag{3}$$

where $A_{df}^r$ represents the radiative rate from the states $d$ to $f$. In the summations, $i'$ and $f'$ run over all the possible autoionization and radiative final states from the state $d$, respectively. $A_{df}^r$ is given by [27]

$$A_{df}^r = \frac{4\omega^3 \alpha^3}{3g_d} \left| \left\langle \psi_f \left\| T^{(1)} \right\| \psi_d \right\rangle \right|^2 \tag{4}$$

where $\omega$ is the frequency of the decay photons, $\alpha$ the fine structure constant, $T^{(1)}$ is the electronic dipole operator, and $\psi_d = \sum\limits_{\nu} b_{d\nu}\Phi_{\nu}$ and $\psi_f = \sum\limits_{\mu} b_{f\mu}\Phi_{\mu}$ describe the atomic state functions (ASFs) given by mixing the configuration state functions (CSFs) with same symmetries for the states $d$ and $f$, respectively. Here, the CSFs $\Phi$ are antisymmetric sums of the products of $N$ one-electron Dirac spinors. The autoionization rate can be given by [17]

$$A_{di}^a = 2\sum\limits_{\kappa} \left| \left\langle \psi_i, \kappa; J_T M_T \left\| \sum\limits_{p<q} (V_{\text{Coul}} + V_{\text{Breit}}) \right\| \psi_d \right\rangle \right|^2 \tag{5}$$

Similar to the previous equation, here, $\psi_i$ represents the ASFs of the autoionizing state; $\kappa$ is the relativistic angular quantum number that describes the free electron; $J_T M_T$ are the total angular momentum of the coupled system, that is, "the target ion plus the impacting electron"; $V_{\text{Coul}} = 1/r_{pq}$ denotes the Coulomb operator of the electron–electron interaction; and $V_{\text{Breit}}$ is the Breit operator [28,29].

For a given resonance state, the strengths $S_{id}$ can be obtained by summing the individual resonance strength $S_{idf}$ over all possible final states $f$:

$$S_{id} = \sum\limits_{f} S_{idf} \tag{6}$$

To compare the theoretical calculations with experimental results, such as those measured in EBIT, the calculated cross-section should be convoluted with a Gaussian distribution describing the experimental electron beam's energy resolution. Thus, the total $DR$ cross-sections from an initial level $i$ can be obtained by summing the contributions of $DR$ strength $S_{id}$ over all resonance states $d$ [27,30]:

$$\sigma_{tot}^{DR}(\varepsilon) = \sum\limits_{d} \frac{2}{\Delta E} \left(\frac{\ln 2}{\pi}\right)^{1/2} \exp\left[-4\ln 2\left(\frac{\varepsilon - E_{id}}{\Delta E}\right)^2\right] S_{id} \tag{7}$$

where $\Delta E$ is the full width at half maximum (FWHM).

In plasma modeling, the $DR$ rate coefficients are often needed. Suppose that the electron velocity is in the Maxwellian distribution; thus, the $DR$ rate coefficients $\alpha_{id}^{DR}$ can be expressed as [30]

$$\alpha_{id}^{DR}(T_e) = \frac{4E_{id}}{\left(2\pi m_e k_B^3 T_e^3\right)^{1/2}} \exp\left(-\frac{E_{id}}{k_B T_e}\right) S_{id} \tag{8}$$

where $T_e$ is the electron temperature and $k_B$ is the Boltzmann constant.

## 3. Results and Discussion

We performed level-by-level computations to obtain the recombination cross-sections and rate coefficients of Mg-like $Au^{67+}$ ions. In the calculations, to explore the effects

of electron correlation on resonance energies and strengths, two electron correlation models, labeled as A and B, were adopted. In model A, the important Al-like double-excited configuration complexes $2p^{-1}3s^23lnl'$ and $2s^{-1}3s^23lnl'$ ($l = p$, $d$ and $n$ = 3–12, $l' \leq 5$) were considered (in which $-1$ indicates the hole); that is, only *DR* channels were included (see Equation (1) (*a*)), which comprise 11,437 resonant states. Model B included both the resonant double-excited configurations in model A and the resonant triple-excited configuration complexes $2p^{-1}3s^{-1}3l^2nl'$ and $2s^{-1}3s^{-1}3l^2nl'$ ($l = p$, $d$ and $n$ = 3–12, $l' \leq 5$); that is, both the *DR* and *TR* channels were included (see Equation (1) (*b*)), which comprise 147,706 resonant states. The radiative and autoionization final states considered in the present resonance recombination calculations are the same in both model A and model B. Specifically, we included the configurations $3s^2nl'$, $3s3lnl'$ and $3l^2nl'$ ($l = p$, $d$ and $n$ = 3–12, $l' \leq 5$) for the radiative final states of Al-like Au$^{66+}$ ions, which means that dominant valence–valence, core–valence, and core–core correlations were incorporated. For autoionizing final states, both the ground configuration $2s^22p^63s^2$ and the lower single- and double-excited configurations $2s^22p^63lnl'$ ($l$ = $s$, $p$, $d$ and $n$ = 3–5, $l' \leq n-1$) of Mg-like Au$^{67+}$ ions were included. Using the FAC code (Version 1.1.5) proposed by Gu [17], which included Berit interactions in the zero-energy limit for the exchanged photons, and quantum electrodynamic (QED) effects [31], we carefully calculated the energy levels and radiative and autoionization rates of Al-like Au$^{66+}$ ions. Note that configuration mixing between all the radiative final states and the resonant excited states from *DR* and *TR* were included in the calculations. Figure 1 illustrates the energy levels of the Mg-like Au$^{67+}$ and Al-like Au$^{66+}$ ions calculated in model B. In Table 1, we compare the energies of Al-like Au$^{66+}$ ions calculated in models A and B as well as with the available theoretical results from Refs. [23,24]. It is found that for the lowest 34 levels (Label 1–34) associated with the $3s^23p$, $3s3p^2$, $3s3p3d$, $3p^23d$ and $3p^3$ configurations, the energy levels given by model B show small deviations from the results of model A within about 1.0 eV. The results in model B are closer to the results calculated by Vilkas et al. [23] using the Multi-Reference Møller–Plesset (MRMP) method and also are in good agreement with the FAC results obtained by Hamasha et al. [24] with differences of less than 0.18% and 0.15%, respectively. However, for the resonant excited states (Label 2862–3819), the energy levels given by model B show bigger deviations from the results of model A. Taking the resonant double-excited state $[2p^{-1}_{1/2}(3d^2_{3/2})_2]_{5/2}$ as an example, the energy is 12,858.35 eV in model A, while it increased to 12,884.54 eV in model B, a difference of 26.19 eV. This indicates that the configuration interactions resulting from taking the *TR* channels into account have small effects on the lower excited states (radiative final states) but significant effects on the resonance states of the Al-like Au$^{66+}$ ions.

In Figure 2, we plot the recombination cross-sections of the LMM processes obtained in model A and B. Here, a Gaussian FWHM of 59 eV is used in the calculations, consistent with the experimental *DR* measurements for Ne-like to Al-like gold ions reported by Schneider et al. [15]. Compared to the results in model A, it is obvious that the cross-sections in model B are dramatically reduced, especially for the resonances near the resonance energies of 3.24, 4.25, 4.93 and 5.10 keV, which originate from the resonance-excited states $[2p^{-1}_{3/2}3d_{3/2}3d_{5/2}]_J$, $[2p^{-1}_{1/2}3p_{1/2}3d_{3/2}]_J$, $[2s^{-1}_{1/2}3p_{1/2}3d_{5/2}]_J$ and $[2p^{-1}_{1/2}3d_{3/2}3d_{5/2}]_J$, respectively. This indicates that configuration interactions (CI) between resonances associated with the *DR* and *TR* channels strongly affected (reduce) the resonance strengths or recombination cross-sections of Mg-like gold. Based on model B, an extended calculation has also been conducted considering more resonance-excited configuration complexes $(2s2p)^{-1}3lnl'$ and $(2s2p)^{-1}3s^{-1}3l^2nl'$ ($n$ = 3–12) with $l'$ up to $n-1$. We note that this calculation causes very small changes to the total cross-sections. Thus, we employed model B to calculate the resonance recombination strengths, cross-sections and rate coefficients of Mg-like gold.

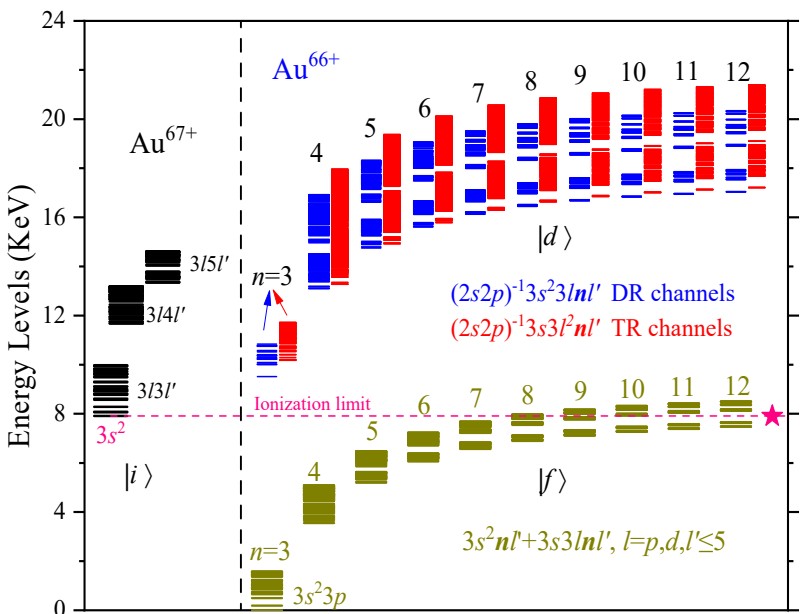

**Figure 1.** The energy level diagram of Mg-like Au$^{67+}$ ions (left) and Al-like Au$^{66+ ions}$ (right) formed by resonant recombination process. $|i\rangle$, $|d\rangle$ and $|f\rangle$ denote the autoionization final states (black), resonance-excited states in *DR* channels (blue) and *TR* channels (red) and the radiation final states (dark-yellow) of the inner L-shell $\Delta n = 1$ resonant recombination process of Mg-like Au$^{67+}$ ions. The pink horizontal dashed line indicates the calculated ionization limit 7910.94 eV of Au$^{67+}$ [3s$^2$], which shows good agreement with the NIST [32] value of 7910 eV (asterisk).

**Table 1.** The calculated energies for the lowest 34 levels and parts of resonance states of Al-like Au$^{66+}$ ions relative to the ground state $[3s^23p_{1/2}]_{1/2}$ with the use of models A and B compared to the MRMP results [23] and FAC results [24].

| Label | Level | P | 2J | This Work | | E$_{MRMP}$ [23] | E$_{FAC}$ [24] |
| | | | | Model A | Model B | | |
|---|---|---|---|---|---|---|---|
| 0 | $[3s^23p_{1/2}]_{1/2}$ | 1 | 1 | 0.00 | 0.00 | 0.00 | 0.00 |
| 1 | $[3s_{1/2}(3p^2_{1/2})_0]_{1/2}$ | 0 | 1 | 187.30 | 186.88 | 186.55 | 186.59 |
| 2 | $[3s^23p_{3/2}]_{3/2}$ | 1 | 3 | 498.12 | 498.13 | 497.56 | 497.92 |
| 3 | $[(3s_{1/2}3p_{1/2})_03p_{3/2}]_{3/2}$ | 0 | 3 | 651.39 | 651.11 | 650.59 | 650.47 |
| 4 | $[(3s_{1/2}3p_{1/2})_13p_{3/2}]_{5/2}$ | 0 | 5 | 671.60 | 671.27 | 670.55 | 670.72 |
| 5 | $[(3s_{1/2}3p_{1/2})_13p_{3/2}]_{3/2}$ | 0 | 3 | 693.58 | 693.22 | 692.65 | 692.80 |
| 6 | $[(3s_{1/2}3p_{1/2})_13p_{3/2}]_{1/2}$ | 0 | 1 | 705.64 | 705.04 | 703.97 | 704.87 |
| 7 | $[3s^23d_{3/2}]_{3/2}$ | 0 | 3 | 753.39 | 753.22 | 752.41 | 753.34 |
| 8 | $[3s^23d_{5/2}]_{5/2}$ | 0 | 5 | 853.55 | 853.56 | 853.25 | 853.16 |
| 9 | $[(3s_{1/2}3p_{1/2})_03d_{3/2}]_{3/2}$ | 1 | 3 | 874.47 | 874.03 | 873.93 | 873.24 |
| 10 | $[(3p^2_{1/2})_03p_{3/2}]_{3/2}$ | 1 | 3 | 888.73 | 888.17 | 887.10 | 887.50 |
| 11 | $[(3s_{1/2}3p_{1/2})_13d_{3/2}]_{5/2}$ | 1 | 5 | 900.18 | 899.62 | 899.32 | 898.92 |
| 12 | $[(3s_{1/2}3p_{1/2})_13d_{3/2}]_{1/2}$ | 1 | 1 | 929.65 | 928.98 | 928.40 | 928.72 |
| 13 | $[(3s_{1/2}3p_{1/2})_13d_{3/2}]_{3/2}$ | 1 | 3 | 938.51 | 937.75 | 936.90 | 937.52 |
| 14 | $[(3s_{1/2}3p_{1/2})_03d_{5/2}]_{5/2}$ | 1 | 5 | 1011.01 | 1010.55 | 1010.34 | 1009.69 |
| 15 | $[(3s_{1/2}3p_{1/2})_13d_{5/2}]_{7/2}$ | 1 | 7 | 1027.34 | 1026.79 | 1026.38 | 1026.02 |
| 16 | $[(3s_{1/2}3p_{1/2})_13d_{5/2}]_{5/2}$ | 1 | 5 | 1038.59 | 1037.87 | 1037.29 | 1037.13 |
| 17 | $[(3s_{1/2}3p_{1/2})_13d_{5/2}]_{3/2}$ | 1 | 3 | 1043.58 | 1042.73 | 1042.01 | 1042.10 |
| 18 | $[(3p^2_{1/2})_03d_{3/2}]_{3/2}$ | 0 | 3 | 1108.54 | 1107.64 | 1106.72 | 1106.85 |
| 19 | $[3s_{1/2}(3p^2_{3/2})_2]_{5/2}$ | 0 | 5 | 1162.12 | 1161.84 | 1160.57 | 1161.11 |
| 20 | $[3s_{1/2}(3p^2_{3/2})_0]_{1/2}$ | 0 | 1 | 1203.57 | 1203.16 | 1201.43 | 1202.58 |
| 21 | $[3s_{1/2}(3p^2_{3/2})_2]_{3/2}$ | 0 | 3 | 1211.30 | 1210.73 | 1209.00 | 1210.42 |
| 22 | $[(3p^2_{1/2})_03d_{5/2}]_{5/2}$ | 0 | 5 | 1236.64 | 1235.60 | 1234.60 | 1234.74 |
| 23 | $[3p_{1/2}(3p^2_{3/2})_2]_{5/2}$ | 1 | 5 | 1366.07 | 1365.57 | 1364.44 | 1364.61 |
| 24 | $[3p_{1/2}(3p^2_{3/2})_2]_{3/2}$ | 1 | 3 | 1372.71 | 1371.98 | 1370.40 | 1371.25 |
| 25 | $[3p_{1/2}(3p^2_{3/2})_0]_{1/2}$ | 1 | 1 | 1391.08 | 1390.52 | 1389.35 | 1389.68 |

**Table 1.** *Cont.*

| Label | Level | P | 2J | This Work | | $E_{\text{MRMP}}$ [23] | $E_{\text{FAC}}$ [24] |
|---|---|---|---|---|---|---|---|
| | | | | Model A | Model B | | |
| 26 | $[(3s_{1/2}3p_{3/2})_2 3d_{3/2}]_{3/2}$ | 1 | 3 | 1398.52 | 1398.00 | 1397.05 | 1397.30 |
| 27 | $[(3s_{1/2}3p_{3/2})_2 3d_{3/2}]_{1/2}$ | 1 | 1 | 1400.65 | 1400.21 | 1398.81 | 1399.45 |
| 28 | $[(3s_{1/2}3p_{3/2})_2 3d_{3/2}]_{5/2}$ | 1 | 5 | 1404.78 | 1404.08 | 1402.80 | 1403.45 |
| 29 | $[(3s_{1/2}3p_{3/2})_2 3d_{3/2}]_{7/2}$ | 1 | 7 | 1406.40 | 1405.83 | 1404.84 | 1405.08 |
| 30 | $[(3s_{1/2}3p_{3/2})_1 3d_{3/2}]_{3/2}$ | 1 | 3 | 1438.22 | 1437.36 | 1436.11 | 1436.97 |
| 31 | $[(3s_{1/2}3p_{3/2})_1 3d_{3/2}]_{5/2}$ | 1 | 5 | 1451.57 | 1450.59 | 1448.92 | 1450.39 |
| 32 | $[(3s_{1/2}3p_{3/2})_1 3d_{3/2}]_{1/2}$ | 1 | 1 | 1458.44 | 1457.30 | 1455.55 | 1457.12 |
| 33 | $[(3s_{1/2}3p_{3/2})_1 3d_{5/2}]_{9/2}$ | 1 | 9 | 1486.85 | 1486.47 | 1485.85 | 1485.18 |
| 34 | $[(3s_{1/2}3p_{3/2})_2 3d_{5/2}]_{5/2}$ | 1 | 5 | 1511.58 | 1511.07 | 1510.17 | 1510.04 |
| 2862 | $[2p^{-1}_{3/2}(3p^2_{3/2})_0]_{3/2}$ | 1 | 3 | 10,550.33 | 10,546.42 | | |
| 3020 | $[(2p^{-1}_{3/2}3d_{3/2})_1 3d_{5/2}]_{7/2}$ | 1 | 7 | 11,098.31 | 11,100.03 | | |
| 3027 | $[(2p^{-1}_{3/2}3d_{3/2})_1 3d_{5/2}]_{5/2}$ | 1 | 5 | 11,105.31 | 11,109.64 | | |
| 3031 | $[(2p^{-1}_{3/2}3d_{3/2})_3 3d_{5/2}]_{7/2}$ | 1 | 7 | 11,106.17 | 11,117.51 | | |
| 3043 | $[(2p^{-1}_{3/2}3d_{3/2})_2 3d_{5/2}]_{5/2}$ | 1 | 5 | 11,122.38 | 11,136.88 | | |
| 3723 | $[(2s^{-1}_{1/2}3p_{1/2})_1 3d_{5/2}]_{7/2}$ | 1 | 7 | 12,801.01 | 12,794.46 | | |
| 3763 | $[2p^{-1}_{1/2}(3d^2_{3/2})_2]_{5/2}$ | 1 | 5 | 12,858.35 | 12,884.54 | | |
| 3819 | $[(2p^{-1}_{1/2}3d_{3/2})_1 3d_{5/2}]_{7/2}$ | 1 | 7 | 13,002.84 | 13,015.12 | | |

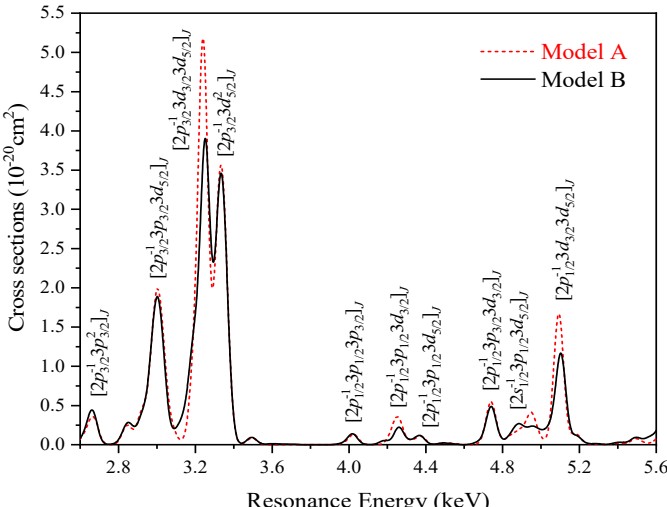

**Figure 2.** The LMM recombination cross-section of Mg-like Au$^{67+}$ ions in the initial ground state, which are obtained by convolving the calculated total strength with a Gaussian profile of 59 eV. The red dotted and black solid lines indicate the results calculated in model A and model B, respectively.

### 3.1. Resonance Energy and Strength

In Table 2, the calculated resonance energies, Auger rate $A^a$, total width $\Gamma_d = \sum A^a + \sum A^r$, radiative branching ratio $B^r_d = \sum_f B^r_{d,f}$ and the resonance strengths corresponding to each of the individual Al-like resonance states formed by *DR* and *TR* processes of Mg-like Au$^{67+}$ ions in the initial ground state are tabulated, where only strong resonances with strengths $S_{id} > 4.0 \times 10^{-20}$ cm$^2$ eV are listed. It was found that the strength of LMM resonances is dominant, which are mainly produced by Mg-like Au$^{67+}$ *DR* processes through the resonance double-excited configurations $(2s2p)^{-1}3l^2$ located in the 2.6–5.6 keV resonance energy range. The strongest resonance comes from the state $[(2p^{-1}_{3/2}3d_{3/2})_1 3d_{5/2}]_{5/2}$, corresponding to the resonance energy of 3252.3 eV, and the resonance strength of $2.16 \times 10^{-18}$ cm$^2$ eV. Most of the *TR* resonances show small resonance strengths; however, some resonances, such as $[(((2p^{-1}_{3/2}3s^{-1}_{1/2})_1 3p_{1/2})_{1/2} 3p_{3/2})_1 3d_{5/2}]_{5/2}$ corresponding to a resonance energy of 3190.0 eV in the LMM manifold, have stronger strengths of $3.44 \times 10^{-19}$ cm$^2$ eV that even surpass the strength of many *DR* resonances.

**Table 2.** The detailed resonance energies $E_{id}$, Auger rate $A^a$, total width $\Gamma_d$, radiative branching ratio $B^r_d$ and resonance strength for dominant *DR* and *TR* resonances ($S_{id} > 4.0 \times 10^{-20}$ cm² eV) of Mg-like Au$^{67+}$ ions. ($a$ ($b$) denotes $a \times 10^b$.)

| LMn | Intermediate Excited State | $E_{id}$ (eV) | $A^a$ (s$^{-1}$) | $\Gamma_d$ (s$^{-1}$) | $B^r_d$ | $S_{id}$ (10$^{-19}$ cm² eV) |
|---|---|---|---|---|---|---|
| | $[(2p^{-1}_{3/2}3p_{1/2})_2 3p_{3/2}]_{7/2}$ | 2102.6 | 9.94 (12) | 1.70 (14) | 0.88 | 0.83 |
| | $[(2p^{-1}_{3/2}3p_{1/2})_2 3p_{3/2}]_{1/2}$ | 2176.1 | 7.17 (14) | 1.86 (15) | 0.11 | 1.74 |
| | $[(2p^{-1}_{3/2}3p_{1/2})_2 3d_{5/2}]_{9/2}$ | 2446.4 | 1.81 (13) | 1.74 (14) | 0.83 | 1.52 |
| | $[(2p^{-1}_{3/2}3p_{1/2})_2 3d_{5/2}]_{1/2}$ | 2493.9 | 7.03 (13) | 3.88 (15) | 0.96 | 1.34 |
| | $[(2p^{-1}_{3/2}3p_{1/2})_1 3d_{5/2}]_{3/2}$ | 2494.2 | 7.57 (13) | 3.88 (15) | 0.97 | 2.90 |
| | $[2p^{-1}_{3/2}(3p^2_{3/2})_0]_{3/2}$ | 2635.5 | 1.47 (13) | 1.84 (14) | 0.84 | 0.47 |
| | $[2p^{-1}_{3/2}(3p^2_{3/2})_2]_{3/2}$ | 2664.4 | 4.92 (14) | 1.38 (15) | 0.14 | 2.49 |
| | $[(2p^{-1}_{3/2}3p_{3/2})_3 3d_{3/2}]_{9/2}$ | 2841.7 | 1.49 (13) | 1.77 (14) | 0.80 | 1.04 |
| | $[(2p^{-1}_{3/2}3p_{3/2})_0 3d_{3/2}]_{3/2}$ | 2923.4 | 3.93 (14) | 1.58 (15) | 0.14 | 1.90 |
| | $[(2p^{-1}_{3/2}3p_{3/2})_3 3d_{5/2}]_{9/2}$ | 2966.1 | 1.29 (13) | 1.71 (14) | 0.84 | 0.90 |
| | $[(2p^{-1}_{3/2}3p_{3/2})_3 3d_{5/2}]_{5/2}$ | 2980.8 | 8.71 (13) | 1.28 (15) | 0.90 | 3.93 |
| | $[(2p^{-1}_{3/2}3p_{3/2})_2 3d_{5/2}]_{5/2}$ | 3001.6 | 6.79 (13) | 2.58 (15) | 0.96 | 3.21 |
| | $[(2p^{-1}_{3/2}3p_{3/2})_3 3d_{5/2}]_{3/2}$ | 3006.0 | 4.62 (13) | 3.63 (15) | 0.97 | 1.47 |
| | $[(2p^{-1}_{3/2}3p_{3/2})_3 3d_{5/2}]_{1/2}$ | 3010.1 | 1.21 (14) | 3.61 (15) | 0.95 | 1.90 |
| | $[(2p^{-1}_{3/2}3p_{3/2})_0 3d_{5/2}]_{5/2}$ | 3031.3 | 2.67 (14) | 1.63 (15) | 0.27 | 3.48 |
| | $[(((2p^{-1}_{3/2}3s^{-1}_{1/2})_2 3p_{1/2})_{3/2} 3p_{3/2})_1 3d_{5/2}]_{7/2}$ * | 3149.2 | 7.77 (12) | 1.56 (14) | 0.86 | 0.42 |
| | $[(2p^{-1}_{3/2}3d_{3/2})_1 3d_{5/2}]_{7/2}$ | 3189.1 | 2.84 (13) | 7.49 (14) | 0.43 | 0.76 |
| | $[(((2p^{-1}_{3/2}3s^{-1}_{1/2})_1 3p_{1/2})_{1/2} 3p_{3/2})_1 3d_{5/2}]_{5/2}$ * | 3190.0 | 8.22 (13) | 2.73 (15) | 0.90 | 3.44 |
| LMM | $[(2p^{-1}_{3/2}3d_{3/2})_1 3d_{5/2}]_{5/2}$ | 3198.7 | 4.60 (13) | 1.18 (15) | 0.69 | 1.47 |
| | $[(2p^{-1}_{3/2}3d_{3/2})_3 3d_{5/2}]_{7/2}$ | 3206.6 | 1.10 (13) | 1.93 (14) | 0.68 | 0.46 |
| | $[(2p^{-1}_{3/2}3d_{3/2})_3 3d_{5/2}]_{5/2}$ | 3218.0 | 4.80 (13) | 9.54 (14) | 0.31 | 0.68 |
| | $[(2p^{-1}_{3/2}3d_{3/2})_2 3d_{5/2}]_{7/2}$ | 3222.7 | 2.21 (13) | 6.47 (14) | 0.32 | 0.44 |
| | $[(2p^{-1}_{3/2}3d_{3/2})_1 3d_{5/2}]_{5/2}$ | 3252.3 | 5.81 (14) | 4.36 (15) | 0.81 | 21.61 |
| | $[(2p^{-1}_{3/2}3d_{3/2})_3 3d_{5/2}]_{1/2}$ | 3256.4 | 6.99 (13) | 4.05 (15) | 0.96 | 1.02 |
| | $[2p^{-1}_{3/2}(3d^2_{5/2})_2]_{7/2}$ | 3315.7 | 1.97 (14) | 8.30 (14) | 0.73 | 8.58 |
| | $[2p^{-1}_{3/2}(3d^2_{5/2})_4]_{7/2}$ | 3338.5 | 2.40 (14) | 1.86 (15) | 0.85 | 12.08 |
| | $[2p^{-1}_{3/2}(3d^2_{5/2})_4]_{5/2}$ | 3358.0 | 5.89 (13) | 4.53 (15) | 0.97 | 2.53 |
| | $[2p^{-1}_{3/2}(3d^2_{5/2})_0]_{3/2}$ | 3363.5 | 9.25 (13) | 3.16 (15) | 0.95 | 2.59 |
| | $2p^{-1}_{1/2}$ | 3491.0 | 2.49 (14) | 5.99 (14) | 0.15 | 0.52 |
| | $[(2p^{-1}_{1/2}3p_{1/2})_0 3p_{3/2}]_{3/2}$ | 4017.6 | 3.32 (14) | 9.27 (14) | 0.10 | 0.85 |
| | $[(2p^{-1}_{1/2}3p_{1/2})_1 3d_{3/2}]_{1/2}$ | 4256.9 | 3.85 (13) | 1.62 (15) | 0.92 | 0.41 |
| | $[(2p^{-1}_{1/2}3p_{1/2})_0 3d_{3/2}]_{3/2}$ | 4260.7 | 1.29 (14) | 8.93 (14) | 0.32 | 0.96 |
| | $[(2p^{-1}_{1/2}3p_{1/2})_0 3d_{5/2}]_{5/2}$ | 4369.2 | 1.50 (14) | 7.62 (14) | 0.14 | 0.70 |
| | $[(2p^{-1}_{1/2}3p_{3/2})_2 3d_{3/2}]_{5/2}$ | 4741.1 | 8.88 (13) | 1.99 (15) | 0.93 | 2.58 |
| | $[(2s^{-1}_{1/2}3p_{1/2})_1 3d_{5/2}]_{7/2}$ | 4883.5 | 2.12 (13) | 9.67 (14) | 0.60 | 0.51 |
| | $[(((2p^{-1}_{1/2}3s^{-1}_{1/2})_1 3p_{1/2})_{1/2} 3p_{3/2})_1 3d_{3/2}]_{5/2}$ * | 4949.5 | 3.10 (13) | 1.72 (15) | 0.75 | 0.69 |
| | $[2p_{1/2}(3d^2_{3/2})_2]_{5/2}$ | 4973.6 | 3.18 (13) | 5.69 (14) | 0.49 | 0.47 |
| | $[(((2p^{-1}_{1/2}3s^{-1}_{1/2})_1 3p_{1/2})_{1/2} 3p_{3/2})_1 3d_{5/2}]_{7/2}$ * | 5055.5 | 5.28 (13) | 1.09 (15) | 0.40 | 0.83 |
| | $[(2p^{-1}_{1/2}3d_{3/2})_1 3d_{5/2}]_{7/2}$ | 5104.2 | 2.07 (14) | 1.91 (15) | 0.84 | 6.73 |
| LMN | $[(2p^{-1}_{3/2}3p_{1/2})_2 4p_{3/2}]_{1/2}$ | 5496.9 | 1.60 (14) | 5.29 (14) | 0.29 | 0.41 |
| | $[(2p^{-1}_{3/2}3p_{1/2})_2 4d_{5/2}]_{3/2}$ | 5623.1 | 3.09 (13) | 1.43 (15) | 0.96 | 0.52 |
| LMM | $[(2s^{-1}_{1/2}3d_{3/2})_2 3d_{5/2}]_{9/2}$ | 5640.6 | 5.08 (13) | 6.92 (14) | 0.33 | 0.74 |
| | $[(2s^{-1}_{1/2}3d_{3/2})_1 3d_{5/2}]_{7/2}$ | 5653.7 | 6.00 (13) | 5.16 (14) | 0.29 | 0.61 |
| | $[(2p^{-1}_{3/2}3p_{3/2})_3 4p_{3/2}]_{3/2}$ | 6004.1 | 1.08 (14) | 4.14 (14) | 0.39 | 0.69 |
| | $[(2p^{-1}_{3/2}3p_{3/2})_0 4p_{3/2}]_{3/2}$ | 6052.7 | 1.82 (14) | 1.46 (15) | 0.35 | 1.03 |
| | $[(2p^{-1}_{3/2}3p_{3/2})_3 4d_{5/2}]_{5/2}$ | 6121.2 | 1.82 (14) | 3.48 (15) | 0.91 | 0.40 |
| | $[(2p^{-1}_{3/2}3p_{3/2})_0 4d_{3/2}]_{3/2}$ | 6137.9 | 7.77 (13) | 1.45 (15) | 0.46 | 0.57 |
| | $[(2p^{-1}_{3/2}3d_{5/2})_1 4p_{1/2}]_{3/2}$ | 6168.4 | 7.40 (13) | 3.41 (15) | 0.93 | 1.10 |
| | $[(2p^{-1}_{3/2}3p_{3/2})_0 4d_{5/2}]_{5/2}$ | 6188.3 | 5.93 (13) | 1.18 (15) | 0.29 | 0.41 |
| | $[(2p^{-1}_{3/2}3d_{3/2})_2 4p_{3/2}]_{3/2}$ | 6246.4 | 4.89 (13) | 1.68 (15) | 0.85 | 0.66 |
| | $[(2p^{-1}_{3/2}3d_{5/2})_4 4p_{3/2}]_{5/2}$ | 6336.1 | 3.87 (13) | 2.82 (14) | 0.71 | 0.65 |
| | $[(2p^{-1}_{3/2}3d_{5/2})_3 4p_{3/2}]_{5/2}$ | 6349.7 | 6.99 (13) | 5.30 (14) | 0.61 | 1.00 |
| LMN | $[(2p^{-1}_{3/2}3d_{5/2})_4 4d_{3/2}]_{7/2}$ | 6408.9 | 1.67 (13) | 2.05 (14) | 0.85 | 0.44 |
| | $[(2p^{-1}_{3/2}3d_{5/2})_4 4d_{5/2}]_{5/2}$ | 6454.7 | 5.42 (13) | 9.45 (14) | 0.91 | 1.14 |
| | $[(2p^{-1}_{3/2}3d_{5/2})_1 4d_{3/2}]_{5/2}$ | 6458.0 | 2.23 (14) | 3.38 (15) | 0.92 | 4.70 |
| | $[(2p^{-1}_{3/2}3d_{5/2})_4 4d_{5/2}]_{7/2}$ | 6466.8 | 7.58 (13) | 4.78 (15) | 0.81 | 1.87 |
| | $[(((2p^{-1}_{3/2}3s^{-1}_{1/2})_2 3p_{1/2})_{5/2} 3d_{5/2})_5 4p_{3/2}]_{7/2}$ * | 6485.0 | 1.99 (13) | 3.06 (14) | 0.80 | 0.49 |
| | $[(((2p^{-1}_{3/2}3s^{-1}_{1/2})_1 3p_{1/2})_{1/2} 3d_{5/2})_3 4p_{3/2}]_{7/2}$ * | 6497.5 | 1.74 (13) | 1.28 (15) | 0.96 | 0.51 |
| | $[(2p^{-1}_{3/2}3d_{5/2})_1 4d_{5/2}]_{7/2}$ | 6499.9 | 5.72 (13) | 1.83 (15) | 0.95 | 1.65 |
| | $[(2p^{-1}_{3/2}3d_{5/2})_1 4d_{5/2}]_{3/2}$ | 6503.2 | 3.93 (13) | 2.86 (15) | 0.96 | 0.57 |
| | $[(2p^{-1}_{3/2}3d_{5/2})_1 4f_{5/2}]_{7/2}$ | 6573.5 | 1.02 (14) | 3.41 (15) | 0.95 | 2.93 |
| | $[(2p^{-1}_{3/2}3d_{5/2})_1 4f_{7/2}]_{9/2}$ | 6588.0 | 8.27 (13) | 3.13 (15) | 0.96 | 2.97 |
| | $[(2p^{-1}_{3/2}3d_{5/2})_1 4f_{7/2}]_{7/2}$ | 6592.8 | 1.51 (13) | 3.53 (15) | 0.98 | 0.44 |
| | $[(2p^{-1}_{3/2}3d_{5/2})_1 5d_{5/2}]_{7/2}$ | 7901.0 | 3.64 (13) | 3.67 (14) | 0.87 | 0.79 |
| | $[(2p^{-1}_{3/2}3d_{5/2})_1 5d_{3/2}]_{5/2}$ | 7914.0 | 7.20 (13) | 3.20 (15) | 0.96 | 1.30 |
| LMO | $[(2p^{-1}_{3/2}3d_{5/2})_1 5d_{5/2}]_{7/2}$ | 7940.1 | 4.95 (13) | 3.99 (15) | 0.97 | 1.20 |
| | $[(2p^{-1}_{3/2}3d_{5/2})_1 5f_{5/2}]_{7/2}$ | 7972.6 | 7.78 (13) | 3.86 (15) | 0.96 | 1.86 |
| | $[(2p^{-1}_{3/2}3d_{5/2})_1 5f_{7/2}]_{9/2}$ | 7980.4 | 5.49 (13) | 3.81 (15) | 0.97 | 1.65 |
| LMN | $[(2p^{-1}_{1/2}3d_{3/2})_1 4d_{5/2}]_{7/2}$ | 8262.6 | 2.25 (13) | 1.44 (15) | 0.90 | 0.49 |
| | $[(2p^{-1}_{1/2}3d_{5/2})_2 4d_{3/2}]_{7/2}$ | 8299.1 | 8.76 (13) | 1.08 (15) | 0.83 | 1.74 |

**Table 2.** *Cont.*

| LMn | Intermediate Excited State | $E_{id}$ (eV) | $A^a$ (s$^{-1}$) | $\Gamma_d$ (s$^{-1}$) | $B^r_d$ | $S_{id}$ ($10^{-19}$ cm$^2$ eV) |
|---|---|---|---|---|---|---|
|  | $[(2p^{-1}_{3/2}3p_{3/2})_0 6d_{3/2}]_{3/2}$ | 8390.7 | 4.76 (13) | 2.70 (14) | 0.79 | 0.45 |
| LMP | $[(2p^{-1}_{3/2}3d_{5/2})_1 6d_{3/2}]_{5/2}$ | 8701.6 | 5.83 (13) | 3.76 (15) | 0.98 | 0.98 |
|  | $[(2p^{-1}_{3/2}3d_{5/2})_1 6f_{5/2}]_{7/2}$ | 8733.0 | 4.82 (13) | 3.76 (15) | 0.99 | 1.08 |
|  | $[(2p^{-1}_{3/2}3d_{5/2})_1 6f_{7/2}]_{9/2}$ | 8737.9 | 3.28 (13) | 3.76 (15) | 0.99 | 0.92 |
|  | $[(2p^{-1}_{3/2}3d_{5/2})_1 7d_{3/2}]_{5/2}$ | 9171.3 | 3.34 (13) | 3.65 (15) | 0.99 | 0.54 |
| LMQ | $[(2p^{-1}_{3/2}3d_{5/2})_1 7f_{5/2}]_{7/2}$ | 9191.3 | 2.82 (13) | 3.67 (15) | 0.99 | 0.60 |
|  | $[(2p^{-1}_{3/2}3d_{5/2})_1 7f_{7/2}]_{9/2}$ | 9194.4 | 2.04 (13) | 3.74 (15) | 0.99 | 0.55 |
| LMR | $[(2p^{-1}_{3/2}3d_{5/2})_1 8f_{5/2}]_{7/2}$ | 9488.2 | 1.96 (13) | 3.75 (15) | 0.99 | 0.41 |
| LMO | $[(2p^{-1}_{1/2}3d_{3/2})_1 5d_{5/2}]_{7/2}$ | 9697.5 | 2.29 (13) | 1.36 (15) | 0.86 | 0.40 |
|  | $[(2p^{-1}_{1/2}3d_{5/2})_2 5d_{3/2}]_{7/2}$ | 9751.3 | 2.53 (13) | 4.82 (14) | 0.89 | 0.46 |

\* Represents the *TR* resonance states.

Figure 3 illustrates the detailed recombination cross-sections for $\Delta n = 1$ LM$n$ ($n = 3$–12) resonance manifolds of Mg-like Au$^{67+}$ ions via the various resonant excited configuration complexes $(2s2p)^{-1}3lnl'$ and $(2s2p)^{-1}3s^{-1}3l^2nl'$ ($n = 3$–12). As can be seen from the figure, the *DR* + *TR* recombination (in model B) spectra of LM$n$ resonance series show similar features in the resonance peaks. For example, the LMM resonances show the largest cross-sections, and the recombination spectra are distributed at 1.5–6.0 keV energy regions, fed by the resonance excitation configuration complexes $(2s2p)^{-1}3l^2$ and $(2s2p)^{-1}3s^{-1}3l^3$ ($l = p, d$). The higher $(2s2p)^{-1}3lnl'$ and $(2s2p)^{-1}3s^{-1}3l^2nl'$ ($n > 3$) complexes have weaker *DR* peaks, appearing at higher energies as $n$ increases and gradually converging to the $(2s2p)^{-1}3l$ and $(2s2p)^{-1}3s^{-1}3l^2$ Rydberg limit (which is at about 13.5 keV). Each resonant peak contains a number of individual resonances, and their resonance energy positions are shown as colored vertical bars in the figure. They are also listed together with the resonant strengths in Table 2 (only for some stronger resonances). It is obvious that the LM$n$ resonant spectra belonging to different manifolds overlap each other and become more and more serious as $n$ increases. Such overlapping usually leads to difficulty in clearly identifying the total *DR* spectra and makes the calculations difficult because of the strong electron–electron correlation effects. To further demonstrate the accuracy of the present results, Figure 4 displays the present calculated LMM *DR* spectra of Mg-like Au$^{67+}$ ions compared with the experimental measurements of the *DR* spectra for Ne-like to Si-like Au ions obtained by Schneider et al. [15] using the LLNL electron beam ion trap. It can be seen that the *DR* spectra positions of Mg-like Au$^{67+}$ ions in the experiment are predicted well by the theoretical calculations. In our previous work [33], the same electron correlation model, namely, model B, was used to calculate the LM$n$ *DR* spectra of Mg-like Fe$^{14+}$ ions. The theoretical results showed very good agreement with the experimental measurements taken by Lukić et al. [34] at the Heidelberg heavy-ion storage ring TSR.

Figure 5 plots the total cross-section for the resonant recombination of Mg-like Au$^{67+}$ as a function of electron energy. To clearly show the higher-order *TR* resonant effects, we identified all *DR* and *TR* resonances and obtained the cross-sections by convolving the resonant strength with a Gaussian FWHM of 59 eV. It is found that the *DR* cross-sections are obviously greater than the *TR* cross-sections in the whole electron collision energy regions from 2.0 to 12.0 keV. Moreover, the contribution of the *TR* process to the total cross-section is about 13.75%, which cannot be neglected.

### 3.2. Rate Coefficients

In Figure 6, the total and partial rate coefficients of Au$^{67+}$ ions through the various resonant excited configuration complexes $(2s2p)^{-1}3lnl'$ and $(2s2p)^{-1}3s^{-1}3l^2nl'$ ($n = 3$–12) are presented as a function of the electron temperature. It can be seen that the maximum rate coefficients appear at about $T_e = 3.4$ keV, and the LMM *DR* processes are the dominant contributors. The $(2s2p)^{-1}3lnl'$ (*DR*) + $(2s2p)^{-1}3s^{-1}3l^2nl'$ (*TR*) ($n > 3$) complex series (solid curves of different colors) gives the predominant electron–ion resonant recombination contribution. At higher electron temperatures, the electron–ion resonant recombination contributions from the $(2s2p)^{-1}3l4l'$ and $(2s2p)^{-1}3s^{-1}3l^24l'$ configuration complex becomes comparable to those of $(2s2p)^{-1}3l3l'$ and $(2s2p)^{-1}3s^{-1}3l^23l'$. Our calculations show that

when $n \geq 10$, the behavior of the total *DR* rate coefficient follows the $n^{-3}$ scaling law [35]. So, we use the $n^{-3}$ scaling law to extrapolate the rate coefficients for $n = 13$–1000 and obtain the total rate coefficient. The contribution from the higher-order excited resonance groups $n \geq 13$ to the total rate coefficient is about 2.93%. Furthermore, in order to investigate the importance of the *TR* process on the plasma ionization balance and radiation energy, we also plot the rate coefficients from the *TR* process (red dashed curve). It is found that the *TR* process contributes about 12.85% to the total rate coefficients and cannot be neglected.

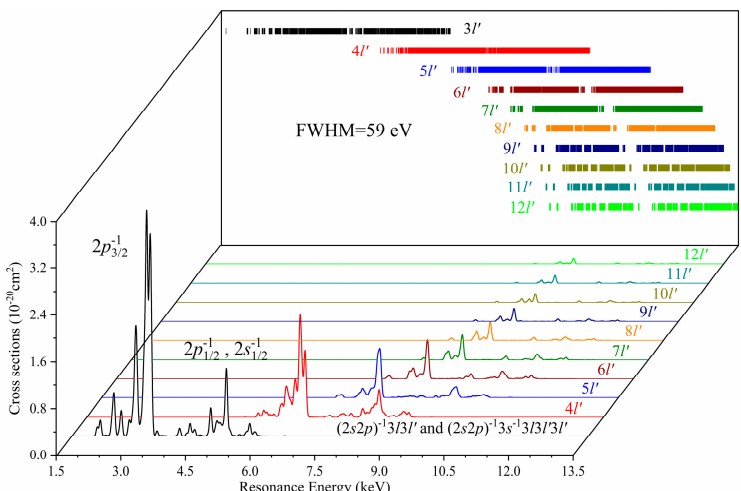

**Figure 3.** Detailed theoretical cross-sections for resonant recombination of Mg-like Au$^{67+}$ through the various Al-like Au$^{66+}$ configuration complexes $(2s2p)^{-1}3lnl'$ and $(2s2p)^{-1}3s^{-1}3l^2nl'$ ($n = 3$–12) as a function of electron energy. A Gaussian FWHM of 59 eV is used. The colored vertical bars show the resonance positions for each of the individual resonances.

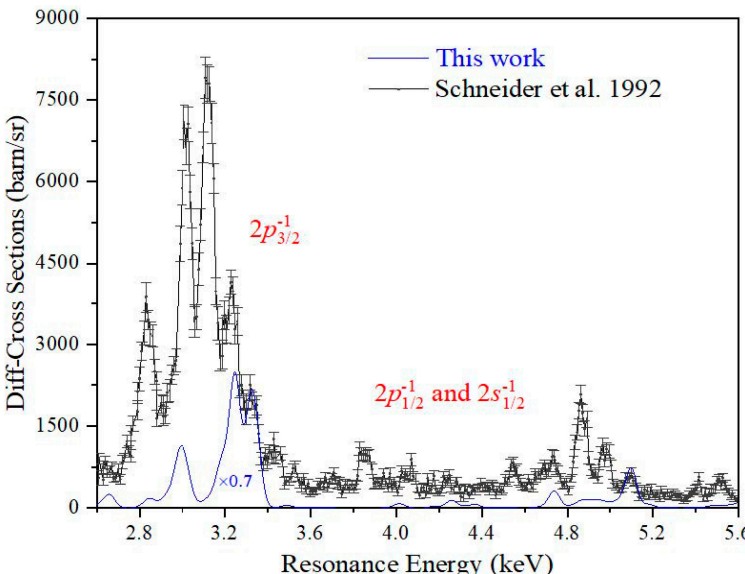

**Figure 4.** The theoretical LMM recombination spectra of Au$^{67+}$ ions multiplied by a factor of 0.7, compared with the EBIT experimental measurements [15]. It should be noted that the experimental results are synthesized LMM spectra of Ne-like to Si-like Au ions.

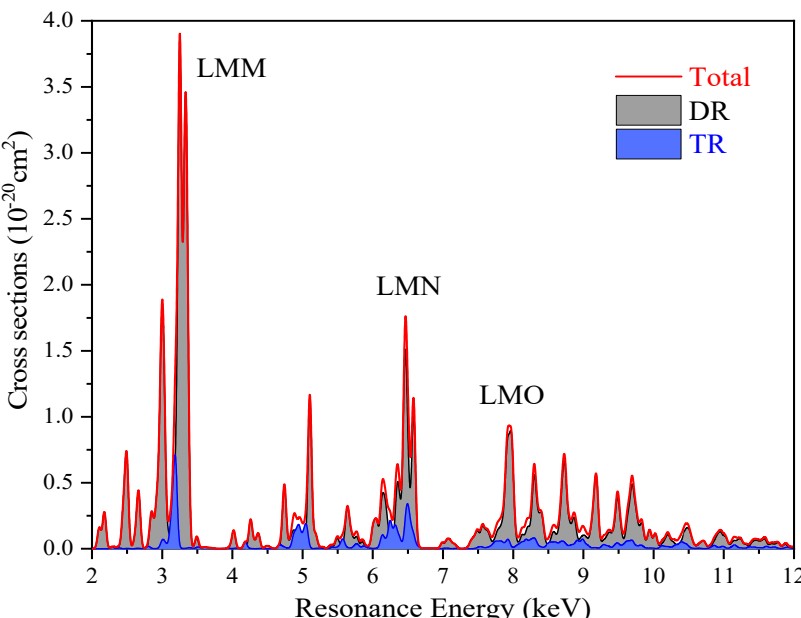

**Figure 5.** The contributions of *DR* and *TR* processes to total cross-section for the $n$ = 3–12 resonance groups of Au$^{67+}$. Dielectronic recombination (*DR*) and trielectronic recombination (*TR*) resonances are identified. The red solid line indicates the total cross-section, and the gray and blue areas show the *DR* and *TR* contributions, respectively.

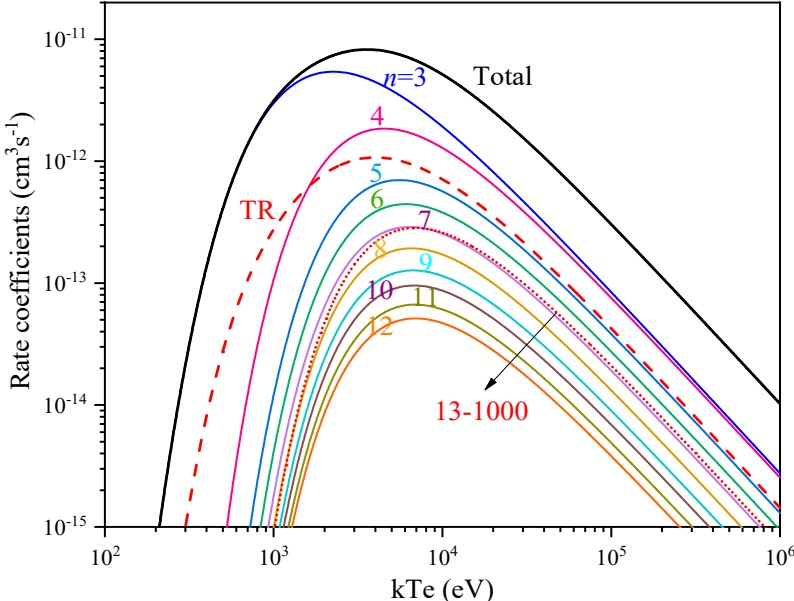

**Figure 6.** The calculated total and partial electron–ion resonant recombination rate coefficients for the $\Delta n$ = 1 excitation resonances of Au$^{67+}$ ions as a function of temperature. The solid curves of different colors represent the contributions of the $(2s2p)^{-1}3lnl'$ and $(2s2p)^{-1}3s^{-1}3l^2nl'$ ($n$ = 3–12) complexes. The dotted curve represents the sum of the contributions of all the $(2s2p)^{-1}3lnl'$ and $(2s2p)^{-1}3s^{-1}3l^2nl'$ ($n$ = 13–1000) complexes. The thick solid line and dashed curve represent the total rate coefficient and the contribution from *TR*, respectively.

For the convenient use of the calculated results in plasma modeling, we fitted the total rate coefficients with the following analytical sum expression [36]:

$$\alpha(T_e) = T_e^{-3/2} \sum_i c_i \times \exp\left(-\frac{E_i}{kT_e}\right) \tag{9}$$

where $k$ is the Boltzmann constant. The fitting parameters of $c_i$ (in cm$^3$ s$^{-1}$ eV$^{3/2}$) and $E_i$ (in eV) are listed in Table 3. Using these parameters in expression (9), it is possible to reproduce the total electron–ion resonant recombination rate coefficients with a good accuracy.

**Table 3.** Fitted parameters $c_i$ (in cm$^3$ s$^{-1}$ eV$^{3/2}$) and $E_i$ (in eV) in Equation (9) for the *DR* rate coefficients of Mg-like Au$^{67+}$ ions. ($a$ ($b$) denotes $a \times 10^b$.)

|  | $c_i$ (cm$^3$ s$^{-1}$ eV$^{3/2}$) | $E_i$ (eV) |
| --- | --- | --- |
| 1 | −2.16 (-6) | 8383.89 |
| 2 | 1.23 (-6) | 11,563.12 |
| 3 | 7.80 (-7) | 3224.54 |
| 4 | 2.56 (-6) | 5934.46 |
| 5 | 6.62 (-6) | 8729.08 |
| 6 | 1.09 (-6) | 3260.14 |
| 7 | 1.94 (-7) | 2362.51 |

## 4. Conclusions

In summary, the $\Delta n = 1$ resonant recombination process associated with $2s/2p$-core excitation of highly charged Au$^{67+}$ ions have been studied theoretically. The resonance energies, resonance strengths, cross-sections and the corresponding electron–ion recombination rate coefficients were carefully calculated for $2s/2p \rightarrow 3l$ transitions using an FAC based on the relativistic configuration interaction method and considering Breit and QED corrections. In the calculations, dominant dielectronic recombination and higher-order trielectronic recombination were considered. It was found that the cross-section of the *DR* process is greater than the *TR* process and that the contribution of the *TR* process to the total cross-section is about 13.75%, which cannot be neglected. Furthermore, the plasma rate coefficients are deduced from the calculated recombination spectra, and the contributions from the higher-order excited resonance groups $n \geq 13$, which were about 2.93% to total rate coefficient, were evaluated using an extrapolation method. It was also found that the *TR* processes are important, which contribute 12.85% of the total rate coefficients.

**Author Contributions:** Formal analysis, L.X. and C.D.; investigation, W.H., S.N. and J.R.; writing original draft preparation, W.H.; writing—review and editing, L.X., C.D. and W.H.; validation, Y.M. All authors have read and agreed to the published version of the manuscript.

**Funding:** The work was supported by the National Key Research and Development Program of China (Grant Nos: 2022YFA1602500, 2017YFA0402300), the Natural Science Foundation of China (Grant Nos: 12064041, 11874051, 12104373), the Innovative Fundamental Research Group Project of Gansu Province (20JR5RA541), and the doctoral research fund of Lanzhou City University (LZCU-BS2019-50).

**Data Availability Statement:** Data in numerical form are available upon request. The fitted coefficients are available in the paper.

**Conflicts of Interest:** The authors declare no conflict of interest.

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
