# Peer review of "Theoretical Investigation of Electron–Ion Recombination Process of Mg-like Gold"

_atoms, doi:10.3390/atoms11050076_

Round 1

Reviewer 1 Report

This paper presents an interesting theoretical investigation of dielectronic recombination (DR) of Mg-like Au. The DR process has a significant effect on the ionization balance of the plasma as well as spectral emission from the plasma, thus this paper is of interest to the community.

Minor amendments may be necessary before publication in Atom. The recombination rates should be calculated with a model, which takes sufficient DR channels into account. The model is presented using tables, but if it is also presented graphically, for example using a level diagram, the suitability of the model would be easier to understand.

Modeling plasmas require a database of atomic data over a wide variety of ions, and the present work would constitute a part of such data. In this paper, the DR process has been investigated based on the established method and also using the reliable atomic code. However, the calculated results still should be validated after comparison with experiments. The authors should show a comparison between calculated and measured cross sections in fig. 2 and 3. Even if experimental data is not available, it would be informative if the authors address possible validation based on present experiments, which have been carried out for a similar system.

Reviewer 2 Report

Refree's report on "Theoretical investigation of electron-ion recombination process of Mg-like gold" by L. Xie et al. 

This paper reports on theoretical results of electron impact recombination for Mg-like gold ion Au^67+.

Here are comments and suggestions.

1. In the last paragraph of the introduction, the authors stated they will focus on the L-shell delta n = 1 DR process of  Au67+ and the DR study for Au67+ is few while the structure and radiative data are many . Did you find neither experimental nor theoretical data for Au67+ to be compared with your results?

Also, are there any works on delta n = 0 DR process of Au67+? or not ?

Please clarify those and mention them in the introduction.

What does RCI(relativistic configuration interaction) mean for FAC? It is very ambiguous terminology compared with RCI used in the codes of MCD(H)F. How different the RCI in fac from RCI in MCD(H)F? The authors need to describe the RCI clearly and briefly elsewhere in their manuscript. Just reference citation for it will not be sufficient.

The reference number ordering [26-27] seems to be error. They should be 23-24. Please check the reference ordering overall in the whole manuscript.

2. In section 2, DId you use atomic units or not? S_id written in Eq. (2) and in Eq. (3) are not consistent with each other. Please check them and mention the unit.

For eq. (5) (line 96), please note for what delta E is the full width at half maximum. Maybe collision energy?  In the reference [25], the strength was expressed with Gamma rather than delta E. What is the relationship of the two?

3. In section 3,  the autioionization and radiative decay channels mentioned (line 111-117) would be better also expressed in Eq. (1) of the section 2 in detail.

Did you include configuration interaction between all resonances from DR and TR (11437+147706 levles) in model B?

As for table 1, the authors stated the radiative and autoionization final states considered in the present DR calcualtions are the same in both mode(mabe typo model?)  A and model B (line 111-112), but why do the calculated energies of levels for Au66+ (radiative channels) depend on the model A and B in the table 1?

4. In section 4, line 235-236, "Breit and QED corrections" is written suddenly in the conclusion firstly. Do you use full Breit correction? And what is QED corrections in the FAC calculation?  

The correlation effects among resonance levels reported here was known also for other ions in other previous works. If there is no available experimental or theoretical work for DR of Au67+, this work will be a useful data for comparison of others in future. However, the issues raised in the above comments need to be resolved.

Therefore,  the acceptance can be reconsidered after revision.
